# Evidence and Future Perspectives for Neoadjuvant Therapy for Resectable and Borderline Resectable Pancreatic Cancer: A Scoping Review

**DOI:** 10.3390/cancers16091632

**Published:** 2024-04-24

**Authors:** Yutaka Endo, Minoru Kitago, Yuko Kitagawa

**Affiliations:** Department of Surgery, Keio University School of Medicine, Shinanomachi 35, Shinjuku, Tokyo 160-8582, Japan; yutakamed91@gmail.com (Y.E.); kitagawa@a3.keio.jp (Y.K.)

**Keywords:** neoadjuvant therapy, pancreatic cancer, randomized controlled trials, scoping review

## Abstract

**Simple Summary:**

Pancreatic cancer presents a challenge due to its high mortality rates and limited treatment options. In an effort to improve patient outcomes, neoadjuvant treatment (NAT) has emerged as a promising approach for both resectable and borderline resectable pancreatic cancer. This scoping review provides a comprehensive overview of evidence from published (n = 14) and ongoing (n = 12) randomized Phase II and III trials, shedding light on the current status of NAT. The efficacy of NAT in terms of survival benefits for patients with resectable pancreatic cancer has been still controversial. However, the efficacy of NAT has been affirmed in cases of borderline resectable pancreatic cancer, although the ideal treatment regimens remain subject to debate. Ongoing trials are actively exploring novel approaches, including immunotherapy, underscoring the dynamic nature of pancreatic cancer treatment. Future efforts aim to refine treatment strategies by integrating systemic chemotherapy with immunotherapy, guided by molecular-based biomarkers to achieve precision oncology.

**Abstract:**

Pancreatic cancer (PC) is a lethal disease that requires innovative therapeutic approaches to enhance the survival outcomes. Neoadjuvant treatment (NAT) has gained attention for resectable and borderline resectable PC, offering improved resection rates and enabling early intervention and patient selection. Several retrospective studies have validated its efficacy. However, previous studies have lacked intention-to-treat analyses and appropriate resectability classifications. Randomized comparative trials may help to enhance the clinical applicability of evidence. Therefore, after searching the MEDLINE database, this scoping review presents a comprehensive summary of the evidence from published (n = 14) and ongoing (n = 12) randomized Phase II and III trials. Diverse regimens and their outcomes were explored for both resectable and borderline resectable PC. While some trials have supported the efficacy of NAT, others have demonstrated no clear survival benefits for patients with resectable PC. The utility of NAT has been confirmed in patients with borderline resectable PC, but the optimal regimens remain debatable. Ongoing trials are investigating novel regimens, including immunotherapy, thereby highlighting the dynamic landscape of PC treatment. Studies should focus on biomarker identification, which may enable precision in oncology. Future endeavors aim to refine treatment strategies, guided by precision oncology.

## 1. Introduction

Pancreatic cancer (PC) is a devastating disease and a major cause of cancer-related death worldwide [1]. Surgically resectable PC accounts for approximately 10–15% of all PC cases; the 5-year overall survival rate for this subset is approximately 25% [2]. Treatment for early-stage PC typically involves a combination of curative-intent pancreatectomy and adjuvant chemotherapy. However, the prognosis following surgical resection with adjuvant chemotherapy, the current standard of care for PC, remains unsatisfactory, with a 5-year overall survival rate of 25–40% [3,4]. Accordingly, the role of neoadjuvant treatment (NAT) in improving the survival outcomes of patients with borderline resectable or locally advanced PC has evolved [5]. The most crucial predictors of survival in patients with PC include surgery with curative intent, early-stage disease (i.e., no lymph node metastases), and complete (R0) resection without distant organ metastases [6]. The efficacy of NAT has been investigated for decades [7,8,9,10]. NAT reportedly enhances the margin-free resection rate [11]. NAT can also function as a valuable tool for patient selection: it aids the identification of individuals who may not benefit from aggressive surgery owing to the presence of highly aggressive disease, thereby helping in the prevention of unnecessary surgeries in such patients [12]. Conversely, NAT has been observed to be correlated with decreased physical function and lower quality of life [13,14]. 

Single-arm studies have been conducted to investigate the safety and efficacy of NAT for PC [15,16]. Early evidence of NAT’s efficacy has been mainly derived from a retrospective analysis of such single-arm studies; with inherent selection bias and the lack of an intention-to-treat (ITT) approach, the findings have limited clinical applicability [17]. Recently, several randomized clinical trials have revealed favorable results regarding the R0 resection rate and subsequent survival [10,18]. Furthermore, the concept of resectability has also been introduced and utilized for patient stratification [17]. Several definitions of PC resectability have been approved for the determination of the possibility of an R0 resection, taking into account oncological aspects and the performance status [9,19]. To enhance the clinical applicability of the existing evidence, analyses targeting randomized controlled trials (RCTs) that consider the classification of resectability are crucial, since the optimal treatment strategies would be different between resectable and borderline resectable PC [19]. In addition, novel treatment approaches, such as immunotherapy and total induction treatment, have been investigated in recent years [20,21].

Therefore, through this review, we aimed to provide a comprehensive summary of the existing evidence from randomized Phase II and III trials focusing on both resectable and borderline resectable PC. We further explored ongoing randomized Phase II and III trials on novel treatments for these cancers. Finally, we aimed to explore potential avenues for future research and offer perspectives on this subject.

## 2. Materials and Methods

A scoping review approach was considered appropriate for this study, in accordance with guidelines published previously, and it adhered to the PRISMA-ScR guidelines [22,23]. On January 15, we comprehensively searched the MEDLINE database for relevant articles published between January 1991 and January 2024. The search terms were as follows: ((pancreatic neoplasm) OR (pancreatic carcinoma) OR ((pancreas*) AND ((cancer*) OR (neoplasm*) OR (adenocarcinoma*) OR (tumor*)))) AND ((neoadjuvant therapy) OR (neoadjuvant*)). Additionally, we explored both MEDLINE and ClinicalTrials.gov to identify published and ongoing clinical trials on patients undergoing NAT for non-metastatic localized PC. The inclusion criteria included full-length articles in English and randomized Phase II trials or RCTs reporting outcome data with defined resectability. The exclusion criteria were non-English publications, articles on studies involving patients with metastasis, reviews and meta-analyses, research letters, single-arm trials, and preclinical studies. The titles and abstracts of the identified articles were screened with respect to the exclusion criteria; articles that met the inclusion criteria were subjected to a full-text review. The literature search and data extraction were performed by two independent investigators (YE and MK). The extraction process was performed independently by a single reviewer. The data extraction process involved systematically reviewing each source and extracting pertinent data points based on predefined criteria and objectives. Additionally, any discrepancies or uncertainties encountered during the extraction process were resolved by MK. A descriptive analysis of each included study was performed, and data on its methods, participants, interventions, and outcomes were analyzed. Outcomes were either presented as originally reported or calculated from the published raw data if possible. The PRISMA-ScR checklist is provided in the Appendix A.

## 3. Results

Initially, 322 and 24 articles were identified following MEDLINE screening and manual searching, respectively (Appendix A). Among these, 26 fulfilled all inclusion criteria and were incorporated into this review; these comprised eight Phase III RCTs, six randomized Phase II trials, four protocols, and eight registered trials. The published studies were categorized based on cancer resectability (resectable PC and borderline resectable PC) and are further discussed below.

### 3.1. Randomized Phase II and III Trials on Resectable PC

To date, nine studies have been conducted on resectable PC (Table 1) [24,25,26,27,28,29,30,31,32]. Palmer et al. explored the efficacy of gemcitabine (GEM) + cisplatin (n = 26) with respect to that of GEM alone (n = 24) in a neoadjuvant setting [27]. They found that the resection rate (70%) and overall survival (OS) were relatively high in the GEM + cisplatin group, with a median survival time (MST) of 15.6 months; however, a direct comparison between the two groups was not performed. Golcher et al. conducted the first randomized Phase II study comparing upfront surgery (n = 33) with neoadjuvant GEM + cisplatin and concurrent radiotherapy (RT; n = 33). Interestingly, the neoadjuvant and upfront groups did not differ significantly in terms of the OS (17.4 vs. 14.4 months, *p* = 0.96) and margin-free (i.e., R0) resection rates (51.5% vs. 48.5%, *p* = 0.31) [25]. PACT-15 was the first multicenter, open-label, Phase II RCT from Italy on resectable PC (n = 88 patients); it revealed that the PEXG regimen (comprising cisplatin, epirubicin, GEM, and capecitabine) extended the median OS to 38.2 months, thereby validating the usefulness of NAT for resectable PC [28]. Two RCTs and one randomized Phase II trial have been conducted to evaluate the utility and safety of NAT in comparison with those of upfront surgery [26,31,32,33]. The PREOPANC-1 study from the Netherlands revealed that despite a small difference in the MST between the NAT and upfront surgery groups, the OS was better in the NAT group than in the upfront surgery group (17.4 vs. 14.4 months, *p* = 0.025) [32,33]. Similarly, the PREP2/JSAP5 study in Japan revealed that, compared with upfront surgery, NAT led to better OS (26.6 vs. 36.7 months, *p* = 0.015) [31]. However, the most recent international randomized Phase II trial, NORPACT-1, did not demonstrate a survival benefit of neoadjuvant FOLFIRINOX when compared with upfront surgery (OS: 25.1 vs. 38.5 months, *p* = 0.05) [26]. Several randomized Phase II trials have been conducted recently to elucidate the optimal NAT regimens for cohorts with resectable PC [29,30,34]. Notably, no trials have shown significant differences among the suggested regimens. For instance, in the SWOGS 1505 trial, the perioperative FOLFIRINOX group and the perioperative GEM + nab-paclitaxel (PTX) group exhibited similar OS (23.2 vs. 23.6 months) [30]. Another randomized Phase II study, the NEONAX trial, demonstrated that the OS was better with GEM + nab-PTX than with GEM alone (25.5 vs. 16.9 months); however, the authors did not mention whether this difference was significant [29].

### 3.2. Phase II and III Trials for Borderline Resectable PC

Five clinical trials examined the efficacy of NAT for borderline resectable PC (Table 2) [35,36,37,38,39]. Jang et al. first demonstrated the efficacy of NAT for borderline resectable PC in South Korea. Overall, 58 patients were randomized to either receive neoadjuvant chemoradiation (NACRT) with GEM or undergo upfront surgery. An ITT analysis revealed that the 2-year survival (i.e., the primary endpoint of the study) was significantly better in the NACRT group than in the upfront surgery group (40.7% vs. 26.1%; hazard ratio [HR]: 1.97, 95% confidence interval [CI]: 1.07–3.62, *p* = 0.028). In the NUPAT-01 study, Yamaguchi et al. found no significant differences in the R0 resection rate and 3-year OS between patients treated with neoadjuvant mFOLFIRINOX and those treated with GEM + nab-PTX. The Alliance A021501 Phase II randomized clinical trial on borderline PC compared the outcomes of eight cycles of mFOLFIRINOX and seven cycles of mFOLFIRINOX followed by RT (stereotactic body RT at 33–40 Gy in five fractions or hypofractionated image-guided RT at 25 Gy in five fractions). In the first 30 patients, the mFOLFIRINOX with hypofractionated RT arm was closed because the R0 resection rate (33%) was less than the threshold value (40%). The 18-month OS rates were 66.7% and 47.3% in the chemotherapy and chemotherapy + RT arms, respectively. Algenpantucel-L (HyperAcute-Pancreas algenpantucel-L [HAP-a]) is a cancer vaccine comprising allogeneic pancreatic cancer cells. Hewitt et al. compared the efficacy and safety of a combination of standard-of-care chemotherapy (i.e., FOLFIRINOX or GEM + nab-PTX) and HAP-a with those of standard-of-care chemotherapy followed by 5-FU chemoradiation for borderline resectable and locally advanced PC [36]. Their study findings did not indicate a superior survival benefit of HAP-a (resection rate, 23% vs. 26%; MST, 14.3 months vs. 14.9 months). In the ESPAC-5 study, patients with borderline resectable PC were randomly assigned to receive immediate surgery (arm 1), neoadjuvant GEM + capecitabine (arm 2), FOLFIRINOX (arm 3), or capecitabine + RT (arm 4) in a 1:1:1:1 allocation ratio [35]. The resection rate was higher but the R0 resection rate was lower in the immediate surgery group than in the other neoadjuvant groups combined (resection rate, 68% vs. 55%; R0 resection rate, 14% vs. 23%). The survival outcome (1-year OS) was better in arms 2 and 3 than in arm 1 (arms 1, 2, 3, and 4: 39%, 78%, 84%, and 60%, respectively; *p* = 0.028). 

### 3.3. Ongoing Trials

Our search revealed 12 ongoing trials on resectable and borderline resectable PC (Table 3) [40,41,42,43,44,45,46,47,48,49,50]. Among these, three aim to assess the benefits of perioperative mFOLFIRINOX. In the Alliance A021806 trial (United States), the researchers aim to compare the outcomes of eight cycles of neoadjuvant mFOLFIRINOX and four cycles of adjuvant mFOLFIRINOX with the outcomes of the standard 12 cycles of adjuvant mFOLFIRINOX for resectable PC [48]. The PREOPANC-3 trial in The Netherlands has followed a similar protocol, comparing the outcomes of eight cycles of neoadjuvant mFOLFIRINOX and four cycles of adjuvant mFOLFIRINOX with the outcomes of the standard 12 cycles of adjuvant mFOLFIRINOX for resectable PDAC [41]. In both trials, the OS is considered the primary endpoint. In the NeoFOL-R trial (South Korea), the researchers aim to compare the efficacy of six cycles of neoadjuvant mFOLFIRINOX and six cycles of adjuvant mFOLFIRINOX with that of the standard 12 cycles of mFOLFIRINOX for resectable PDAC, with the 2-year survival rate being the primary outcome [45]. The PANACHE01-PRODIGE48 is a three-arm, non-comparable, randomized Phase II trial on resectable PC; patients have been allocated (2:2:1) to arms A (four cycles of neoadjuvant mFOLFIRINOX and eight cycles of adjuvant chemotherapy), B (four cycles of neoadjuvant FOLFOX and eight cycles of adjuvant chemotherapy), and C (upfront surgery followed by 12 cycles of adjuvant chemotherapy) [40]. In the PANDAS-PRODIGE 44 trial, 90 patients with borderline resectable PC will receive neoadjuvant mFOLFIRINOX with (arm A) or without (arm B) concurrent RT (50.4 Gy), with both regimens followed by surgery and adjuvant therapy [44]; the primary endpoint of this trial is the R0 resection rate. Three studies are investigating GEM-based neoadjuvant regimens. In the CSPAC-28 trial, patients with borderline resectable PC have been randomly allocated to arms A (GEM + nab-PTX) and B (mFOLFIRINOX); the primary endpoint is the OS [42]. The researchers in the CISPD-1 trial aim to compare sequential GEM + nab-PTX and mFOLFIRINOX administration with upfront surgery in patients with resectable PC; the primary endpoint is the disease-free survival (DFS) [46]. Additionally, the researchers from a trial from Japan aim to assess the efficacy of neoadjuvant GEM + S-1 in comparison with that of GEM + nab-PTX for resectable or borderline resectable PC; the primary endpoint is the progression-free survival [49]. Two studies have focused on recently developed systemic treatments. The CAPT-02 study is focused on the utility of NALIRIFOX (liposomal irinotecan + oxaliplatin + 5-FU/LV) with or without adebellizumab in a neoadjuvant setting [43]. The researchers from the other study, a randomized Phase II study, aim to compare the outcomes of neoadjuvant pembrolizumab + defactinib with those of neoadjuvant pembrolizumab; the primary endpoint is the pathological response rate [47]. Appendix A depict the flow diagrams for the representative trials, Alliance A021806 and PANACE01-PROGIGE 48.

## 4. Discussion

Despite advances in perioperative treatment, PC remains a significant health burden and a leading cause of cancer-related mortality [1]. NAT offers several potential benefits, including increased chances of achieving margin-negative resection and assistance with the selection of oncological patients [5]. An exploration into NAT as a treatment modality for patients with resectable or borderline resectable PC has been a focal point in enhancing survival outcomes [3]. However, the efficacy and safety of NAT in these patient populations are not completely elucidated. Alongside investigations aimed at verifying the effectiveness of NAT, studies have also focused on recently developed systemic and adjuvant chemotherapy regimens in order to determine an optimal treatment regimen [4,52,53,54,55]. Despite these efforts, a definitive and effective treatment regimen has not yet been established. The present scoping review is important because it provides a comprehensive overview of randomized Phase II and III clinical trials, while incorporating the definition of resectability criteria. It covers ongoing clinical trials that could impact the current guidelines, thereby presenting a valuable resource for an understanding of the landscape of PC treatment and the anticipation of potential paradigm shifts [19,56,57].

The rationale for the recommendation of NAT for potentially resectable PC is based on the following key advantages that it offers: (1) the suppression of primary tumors and elimination of potential micrometastases that may not be visible during preoperative imaging; (2) the reduction of the tumor volume and an increase in the R0 resection rate; (3) the screening of biological behaviors, allowing for individualized treatment [12,58]. Nonetheless, the optimal treatment approach for resectable PC remains controversial [18]. For instance, through an ITT analysis, Unno et al. demonstrated that neoadjuvant GEM + S-1 treatment resulted in better survival than upfront surgery; although their study is yet to be published, their findings support the use of NAT for resectable PC [31]. Similarly, the PREOPANC-1 trial indicated a survival benefit of neoadjuvant GEM + RT when compared with upfront surgery, albeit with a margin of only 1.4 months [32]. Conversely, the most recent NORPANC-01 trial by Labori et al. demonstrated no survival benefit of neoadjuvant FOLFIRINOX for resectable PC [26]; the authors suggested that this insignificant result may be attributed to the markedly better prognosis in the upfront surgery group (MST: 38.5 months). They proposed that recent advances in imaging modalities and diagnostic capabilities could enhance patient selection, sparing individuals previously deemed to have resectable cancer but actually having unresectable cancer due to distant micrometastases [26]. They also emphasized the importance of avoiding postoperative complications that could hinder the successful completion of adjuvant chemotherapy [26]. To elucidate the survival advantages offered by NAT over upfront surgery, the outcomes of ongoing trials (such as Alliance A021806 and PANACE01-PRODIGE 48) are anticipated. Overall, although NAT is a novel and promising treatment strategy for resectable PC, both its efficacy and optimal regimen are yet to be determined. It must be noted that during NAT, the patients’ physical function and global health assessment results are significantly impaired [59]. Thus, the balance between the advantages and drawbacks of NAT in patients with resectable PC should be carefully monitored, because the available evidence remains inconsistent. Ongoing trials, such as Alliance A021806 and PANACHE-PRODIGE 44, have included quality of life as one of the secondary outcomes (Appendix A) [44,48]; their results are required to attain robust conclusions. 

In recent trials, the effectiveness of NAT for borderline resectable PC has become apparent when compared with that of immediate surgery. Jang et al. demonstrated that, compared with immediate surgery followed by adjuvant GEM, neoadjuvant GEM + RT resulted in a superior R0 resection rate and OS [37]. The ESPAC 05 study similarly revealed that, compared with the immediate surgery group, all NAT groups exhibited improved OS and DFS; notably, the GEM + capecitabine and FOLFIRINOX groups exhibited the best outcomes [35]. Consequently, NAT has been recommended for borderline resectable PC [19,56,57]; the current topic of research in the field is the optimal treatment regimen. The findings from the Alliance A021501 and ESPAC-5 studies suggest that recently developed intensive chemotherapy regimens, such as (m)FOLFIRINOX or GEM + nab-PTX, may be beneficial [35,38]. Thus, the results of ongoing trials investigating the efficacy of such regimens are awaited. Furthermore, several single-arm trials have assessed promising treatment modalities [60,61,62,63,64]. For instance, studies have investigated induction therapy and total NAT (i.e., neoadjuvant systemic chemotherapy followed by short-term chemoradiation) and proven their benefits in terms of an improved resection rate, R0 resection rate, and OS [21,65]. Consequently, further studies are warranted to identify the most efficient treatment regimen.

Recent studies have focused on immunotherapy as a treatment modality for PC [20]; however, despite its success against other cancers, immunotherapy for advanced PC has shown little progress [20,66]. Unlike immunogenic cancers, such as melanomas and small cell lung cancer, advanced PC has not demonstrated a strong response to immune checkpoint inhibitor monotherapy in preclinical or clinical trials [67,68]. A notable clinical trial explored the efficacy of a cancer vaccine (a form of immunotherapy) in a neoadjuvant setting in combination with current systemic chemotherapy [36]. Unfortunately, no significant improvements in the OS (primary endpoint) or DFS (secondary endpoint) were observed. However, ongoing studies are investigating the potential benefits of neoadjuvant immunotherapy, which presents a promising avenue for further exploration [47,69,70]. Another future perspective involves precision oncology based on the patients’ biological backgrounds [71]. Recent studies have underscored the need for more sensitive biomarkers to guide NAT [72]. These biomarkers could help surgeons to identify the optimal subset of patients suitable for NAT. For instance, Nakano et al. found that *KRAS* mutations in postoperative serum samples served as independent prognostic factors for DFS [73]. The combination of liquid biopsy and next-generation sequencing techniques may enhance the accuracy of patient selection [74]. Future research will focus on the development and validation of suitable molecular-based biomarkers.

The interpretation of this review’s findings is subject to several limitations. First, the included trials exhibited heterogeneity, were of a small scale, and served as hypothesis-generating but inconclusive studies. It is important to note that this was not a systematic review; therefore, no formal assessment of the quality of evidence was conducted, and some relevant studies may have been unintentionally excluded. Additionally, the nature of a scoping review precludes the implementation of pooled statistical analyses. To overcome these drawbacks, systematic reviews and meta-analyses should be performed with a relatively similar patient population (such as in terms of cancer resectability and the AJCC stage); furthermore, comprehensive outcomes encompassing efficacy and quality-of-life assessments should be reported.

## 5. Conclusions

In summary, while NAT shows promise as a novel treatment strategy for resectable PC, its efficacy and optimal regimen are not yet fully elucidated. Meanwhile, it is the standard of care for borderline resectable PC. The determination of an optimal treatment regimen remains a topic of ongoing investigation. Future research endeavors will integrate advancements into more potent treatment strategies (combining systemic chemotherapy, RT, and immunotherapy) and molecular-based biomarkers for precision oncology.

## Figures and Tables

**Table 1 cancers-16-01632-t001:** Characteristics of the included studies.

Author	Year	Intervention	Comparator	Primary Outcome	Number of Patients	Resection Rate	R0 Resection Rate	OS
Palmer et al. [27]	2007	GEM + cisplatin	GEM	Resection rate	26 vs. 24	70% vs. 38%	46% vs. 25%	MST: 15.6 mo vs. 9.9 mo
Golcher et al. [25]	2015	GEM + cisplatin with RT (50.4 Gy)	Upfront surgery	OS	33 vs. 33	57.6% vs. 69.7%	51.5% vs 48.5%	MST: 17.4 mo vs. 14.4 mo
Casadei et al. [24]	2015	GEM (2 cycles) followed by combined chemoradiotherapy (6 weeks; 45 Gy and a boost of 9 Gy + GEM)	Upfront surgery	R0 resection	18 vs. 20	61.1% vs. 75.0%	38.9% vs. 25.0%	MST: 22.4 mo vs. 28.3 mo
Reni et al. [28] (PACT-15)	2018	Arm A: Adjuvant GEM Arm B: Adjuvant PEXG Arm C: Neoadjuvant and adjuvant PEXG		1-year event-free survival, DFS, and OS	26 vs. 30 vs. 32	88.0% † vs. 84.4%	28.6% * vs. 53.1%	20.4 mo vs. 26.4 mo vs. 38.2 mo
Unno et al. [31] (PREP2/JSAP05)	2019	GEM + S-1	Upfront surgery	OS	182 vs. 180			36.7 mo vs. 26.6 mo *
Sohal et al. [30] (SWOG S1505)	2021	FOLFIRINOX (3 cycles) with adjuvant FOLFIRINOX (3 cycles)	GEM/nab-PTX (3 cycles) with adjuvant GEM/nab-PTX (3 cycles)	OS	55 vs. 47	73% vs. 70%	61.8% vs. 59.6%	23.2 mo vs. 23.6 mo
Seufferlein et al. [29] (NEONAX)	2022	GEM + nab-PTX	GEM	DFS	59 vs. 59	69% vs. 78%	66% vs. 74%	25.5 mo vs. 16.9 mo
Versteijne et al. [32] (PREOPANC-1)	2023	GEM + RT (36 Gy)	Upfront surgery	OS	119 vs. 127	60.5% vs. 72.4%	41.2% vs. 26.8%	15.7 mo vs. 14.3 mo
Labori et al. [26] (NORPACT-1)	2024	FOLFIRINOX (4 cycles)	Upfront surgery	18-month OS	77 vs. 63	82% vs. 89%	56% vs. 39% *	18-mo OS: 60% vs. 73% * MST: 25.1 mo vs. 38.5 mo HR: 1.52 (1.00–2.33)

GEM, gemcitabine; nab-PTX, nab-paclitaxel; RT, radiotherapy; OS, overall survival; DFS, disease-free survival; MST, median survival time; HR, hazard ratio; R0 resection, complete resection; PEXG, cisplatin, epirubicin, gemcitabine, and capecitabine; mo, month(s). *: statistically significant, †: arms A and B combined.

**Table 2 cancers-16-01632-t002:** Randomized Phase II or Phase III studies on NAT for borderline resectable PC.

Author	Year	Intervention	Comparator	Primary Outcome	Number of Patients	Resection Rate	R0 Resection Rate ITT	OS
Jang et al. [37]	2018	GEM + RT + adjuvant GEM	Upfront surgery + adjuvant GEM	2-year survival	27 vs. 23	63.0% vs. 78.3%	51.8% vs. 26.1% *	MST: 21 mo vs. 12 mo HR: 0.51 (0.27–0.93) *
Yamaguchi et al. [39] (NUPAT 01)	2022	mFOLFIRINOX	GEM + nab-PTX	R0 resection rate	26 vs. 25	88.5% vs. 80.0%	73.1% vs. 56.0%	3-year OS: 55.3% vs. 54.4% HR: 0.95 (0.39–2.29)
Katz et al. [38] (Alliance A021501)	2022	mFOLFIRINOX + adjuvant FOLFOX6	mFOLFIRINOX + RT + adjuvant FOLFOX6	18-month survival	65 vs. 55	49% vs. 35%		18-month OS: 66.7% vs. 47.3% MST: 29.8 mo vs. 17.1 mo
Hewitt et al. [36]	2022	FOLFIRINOX or GEM + nab-PTX + HAPa	FOLFIRINOX or GEM + nab-PTX + 5-FU-based CRT	OS	145 vs. 158	23% vs. 26%		MST: 14.3 mo vs. 14.9 mo HR: 1.02 (0.66–1.58)
Ghaneh et al. [35] (ESPAC 05)	2023	Arm 1: immediate surgery Arm 2: GEM + capecitabine Arm 3: FOLFIRINOX Arm 4: capecitabine + RT		Recruitment rate Resection rate	33 vs. 20 vs. 20 vs. 17	55% † vs. 68%	23% † vs. 14%	1-year OS *: 39% vs. 78% vs. 84% vs. 60%

GEM, gemcitabine; nab-PTX, nab-paclitaxel; HAP-a, HyperAcute-Pancreas; RT, radiotherapy; OS, overall survival; HR: hazard ratio; MST, median survival time; R0 resection, complete resection; FU, fluorouracil; CRT, chemoradiotherapy; mo, month(s); ITT, intention-to-treat. *: statistically significant, †: neoadjuvant groups combined.

**Table 3 cancers-16-01632-t003:** Ongoing trials on the efficacy of NAT for resectable or borderline resectable PC.

Trial Number	Trial Name	Resectability	Intervention	Comparator	Primary Outcome
NCT04340141 [48]	Alliance A021806	R-PC	Perioperative mFOLFIRINOX	Adjuvant mFOLFIRINOX	OS
NCT02959879 [40]	PANACE01-PROGIGE 48	R-PC	Preoperative FOLFOX or mFOLFIRINOX	Upfront surgery	OS
NCT04927780 [41]	PREOPANC-3	R-PC	Perioperative mFOLFIRINOX (8 weeks and 4 weeks)	Adjuvant mFOLFIRINOX (12 weeks)	OS
NCT06172036 [43]	CAPT-02	R-PC	NALIRIFOX with or without adelizumab	Upfront surgery	EFS rate
NCT05529940 [45]	NeoFOL-R	R-PC	Perioperative mFOLFIRINOX (6 weeks and 6 weeks)	Adjuvant mFOLFIRINOX (12 weeks)	2-year survival rate
NCT03750669 [46]	CISPD-1	R-PC	Sequential GEM + nabPTX and mFOLFIRINOX	Upfront surgery	DFS
NCT03727880 [47]	-	R-PC	Neoadjuvant pembrolizumab + defactinib	Pembrolizumab	Pathological response rate
UMIN000021484 [49]	PDAC-GS/GA-rP2, CSGO-HBP-015	R-PC/BR-PC	GEM + S-1	GEM + nab-PTX	PFS
NL7094 [50]	PREOPANC-2	R-PC/BR-PC	Neoadjuvant FOLFIRINOX (8 cycles) without adjuvant treatment	GEM (3 cycles) with RT during the second cycle + adjuvant GEM	OS
NCT02676349 [44]	PANDAS-PRODIGE 44	BR-PC	Neoadjuvant mFOLFIRINOX + CRT (50.4 Gy)	Neoadjuvant mFOLFIRINOX	R0 resection rate
NCT04617821 [42]	-	BR-PC	GEM + nabPTX	mFOLFIRINOX	OS
NCT03777462 [51]	BRPCNCC-1	BR-PC	Neoadjuvant GEM + nab-PTX with SBRT, neoadjuvant S-1 + nab-PTX with SBRT	GEM + nab-PTX	OS

R-PC, resectable pancreatic cancer; BR-PC, borderline resectable pancreatic cancer; SBRT, stereotactic radiotherapy; GEM, gemcitabine; nab-PTX, nab-paclitaxel; OS, overall survival; DFS, disease-free survival; PFS, progression-free survival; EFS, event-free survival; R0 resection, complete resection; RT, radiotherapy.

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
