# Peer review of "Evidence and Future Perspectives for Neoadjuvant Therapy for Resectable and Borderline Resectable Pancreatic Cancer: A Scoping Review"

_cancers, 2024, doi:10.3390/cancers16091632_

Round 1
Reviewer 1 Report
Comments and Suggestions for Authors
The manuscript by Endo et al. provides a thorough overview of evidence from both published (n=14) and ongoing (n=12) randomized Phase II and III trials of neoadjuvant therapy. The review is nicely written and well organized. It will definitely add new insights about the therapy towards PC.
I have a few suggestions for polishing the manuscript.
Comments:
1. The authors should add some more information on Pancreatic Cancer.
2. The authors should state the novelty of NAT compared to other treatments rather than in the Introduction Section.
3. The authors can introduce a flowchart to explain the Materials and Methods.
4. The authors should include a flow diagram explaining all trails at the very end. This will help to understand the manuscript in an easy manner.
5. The authors have stated some limitations at the very end. They should also explain how to overcome these limitations.
Author Response
Comments to the Author
The manuscript by Endo et al. provides a thorough overview of evidence from both published (n=14) and ongoing (n=12) randomized Phase II and III trials of neoadjuvant therapy. The review is nicely written and well organized. It will definitely add new insights about the therapy towards PC.
I have a few suggestions for polishing the manuscript.
Comments:
- The authors should add some more information on Pancreatic Cancer.
Response: Thank you for this suggestion. In accordance with it, we have added additional information about pancreatic cancer (particularly its incidence and associated 5-year overall survival) to the Introduction section as follows (lines 40–43 and 45).
Lines 40–43: “Surgically resectable PC accounts for approximately 10–15% of all PC cases; the 5-year overall survival rate for this subset is approximately 25% [2]. Treatment for early-stage PC typically involves a combination of curative-intent pancreatectomy and adjuvant chemotherapy.”
Line 45: “…with a 5-year overall survival rate of 25–40%”
- The authors should state the novelty of NAT compared to other treatments rather than in the Introduction Section.
Response: Thank you for this suggestion. Accordingly, we have highlighted the novelty of neoadjuvant therapy in comparison with other existing treatment modalities in the Discussion section as follows (lines 221–223).
“NAT offers several potential benefits, including increased chances of achieving margin-negative resection and assistance with selection of oncological patients [5].”
- The authors can introduce a flowchart to explain the Materials and Methods.
Response: Thank you for this comment. Accordingly, we have included the study flowchart as a supplementary figure (see Figure S1, cited on line 102 in the revised manuscript as follows).
“Initially 322 and 24 articles were identified following MEDLINE screening and manual searching, respectively (Figure S1).”
- The authors should include a flow diagram explaining all trials at the very end. This will help to understand the manuscript in an easy manner.
Response: Thank you for this invaluable insight. Accordingly, we have added flow diagrams of some representative trials (namely Alliance A021806 and PANACHE-PRODIGE 44) as supplementary figures (see Figure S2, cited on lines 213 and 262 in the revised manuscript as follows).
Line 213: “Figure S2 depicts a flow diagram of several representative trials.”
Line 262: “Ongoing trials, such as Alliance A021806 and PANACHE-PRODIGE 44, have included quality of life as one of the secondary outcomes (Figure S2)…”
- The authors have stated some limitations at the very end. They should also explain how to overcome these limitations.
Response: Thank you for this suggestion. In accordance with it, we have stated how we believe the listed study limitations can be overcome in the Discussion section as follows (lines 304–308).
“To overcome these drawbacks, systematic reviews and meta-analyses should be performed with a relatively similar patient population (such as in terms of cancer resectability and the AJCC stage); furthermore, comprehensive outcomes encompassing efficacy and quality-of-life assessments should be reported.”

Reviewer 2 Report
Comments and Suggestions for Authors
Well written summary of the current situation of role of neoadjuvant therapy in pancreatic cancer in both routine and locally advanced therapy.
1.The introduction however needs significant revision. In concentrates almost totally on the role in upfront resections and only mentions the role in borderline resectable tumours almost in passing at the end, though it does cover it well in the discussion and methods and results
specific issues with the introduction:
a. line 41. just cant say "unsatisfactory"which means little. Need to actually give some referenced survival data
b. lines 43-45. the most critical predictors is not just early stage disease and R0, but also local invasion of critical structures and distant disease. Patients may well have small tumors without lymph nodes but metastatic disease. This requires revision to aknowledge this
c. lines 46 -47. Suggest that NACV may allow greater margin free resection rate AND downstage tumours locally, The comment about early initiation of treatment preventing deconditioning makes little sense as the treatment may be delayed in resectable cases who have early disease and are not deconditioned whilst the NAC by its aggressive nature may decondition the patient. This balance of treatment, side effects and benefit needs to be nuanced here as well as later in the discussion by referring to need for QoL studies
2. The authors note that OS data for neoadjuvent in the non borderline resectable shows only a small advantage even in those studies which demonstrate an advantage. The authors need to discuss quality of life issues in these patients as toxicity and in hospital stays as part of the whole treatment are high and may lead to deconditioning of the patients and shpuld discuss whether quality of life measure should be a routine measures in these patients and studies in the future
3. The conclusion does not actually give any information apart from the gold statement comment on borderline resectable. In view of the conflicting data on NAC in resectable tumours, perhaps the lines from 248-251 ("Overall, although NAT is a novel and promising treatment strategy for resectable PC, both its efficacy and optimal regimen are yet to be determined") should be repeated or placed there as it provides the most pertinent summary
Comments on the Quality of English Languageno issues with language quality
Author Response
Well written summary of the current situation of role of neoadjuvant therapy in pancreatic cancer in both routine and locally advanced therapy.
1.The introduction however needs significant revision. In concentrates almost totally on the role in upfront resections and only mentions the role in borderline resectable tumours almost in passing at the end, though it does cover it well in the discussion and methods and results
Response: Thank you for your detailed comments on our manuscript. We have addressed all your concerns and have revised our paper accordingly. We are grateful for your insight and hope that our revisions have alleviated your concerns as much as possible.
specific issues with the introduction:
- line 41. just cant say "unsatisfactory"which means little. Need to actually give some referenced survival data
Response: Thank you for flagging this with us. Accordingly, we have provided data on survival in patients with pancreatic cancer following surgical resection and adjuvant chemotherapy, as shown below, and have cited some pertinent references (line 45).
Line 45: “…with a 5-year overall survival rate of 25–40% [3,4]”
References:
- Kleeff, J.; Korc, M.; Apte, M.; La Vecchia, C.; Johnson, C.D.; Biankin, A.V.; Neale, R.E.; Tempero, M.; Tuveson, D.A.; Hruban, R.H.; et al. Pancreatic Cancer. Nat Rev Dis Primers 2016, 2, 16022, doi:10.1038/nrdp.2016.22.
- Uesaka, K.; Boku, N.; Fukutomi, A.; Okamura, Y.; Konishi, M.; Matsumoto, I.; Kaneoka, Y.; Shimizu, Y.; Nakamori, S.; Sakamoto, H.; et al. Adjuvant Chemotherapy of S-1 Versus Gemcitabine for Resected Pancreatic Cancer: A Phase 3, Open-Label, Randomised, Non-Inferiority Trial (JASPAC 01). Lancet 2016, 388, 248-257, doi:10.1016/S0140-6736(16)30583-9.
- lines 43-45. the most critical predictors is not just early stage disease and R0, but also local invasion of critical structures and distant disease. Patients may well have small tumors without lymph nodes but metastatic disease. This requires revision to aknowledge this
Response: Thank you for your invaluable insight. We have modified the flagged sentence to address your concerns accordingly as shown below (line 49).
“The most crucial predictors of survival in patients with PC include surgery with curative intent, early-stage disease (i.e., no lymph node metastases), and complete (R0) resection without distant organ metastases…”
- lines 46 -47. Suggest that NACV may allow greater margin free resection rate AND downstage tumours locally, The comment about early initiation of treatment preventing deconditioning makes little sense as the treatment may be delayed in resectable cases who have early disease and are not deconditioned whilst the NAC by its aggressive nature may decondition the patient. This balance of treatment, side effects and benefit needs to be nuanced here as well as later in the discussion by referring to need for QoL studies
Response: Thank you for flagging this with us. We completely agree with your comment, and have thus, removed the sentence on early initiation of treatment preventing deconditioning. Instead, we have addressed balancing the treatment (side effects versus benefits), especially from the perspective of the quality of life, in the Introduction and Discussion sections as shown below (lines 54–56 and 257–263).
Lines 54–56: “Conversely, NAT has been observed to be correlated with decreased physical function and a lower quality of life [13,14].”
Lines 257–263: “It must be noted that during NAT, the patients’ physical function and global health assessment results are significantly impaired [59]. Thus, the balance between the advantages and drawbacks of NAT in patients with resectable PC should be carefully monitored, because the available evidence remains inconsistent. Ongoing trials, such as Alliance A021806 and PANACHE-PRODIGE 44, have included quality of life as one of the secondary outcomes (Figure S2) [44,48]; their results are required to attain robust conclusions.”
- The authors note that OS data for neoadjuvent in the non-borderline resectable shows only a small advantage even in those studies which demonstrate an advantage. The authors need to discuss quality of life issues in these patients as toxicity and in hospital stays as part of the whole treatment are high and may lead to deconditioning of the patients and shpuld discuss whether quality of life measure should be a routine measures in these patients and studies in the future
Response: Thank you for this insight. As previously mentioned, the drawbacks of neoadjuvant therapy in terms of a deteriorating quality of life have been addressed in the Discussion section as shown below (lines 257–263 and 304–308).
Lines 257–263: “It must be noted that during NAT, the patients’ physical function and global health assessment results are significantly impaired [59]. Thus, the balance between the advantages and drawbacks of NAT in patients with resectable PC should be carefully monitored, because the available evidence remains inconsistent. Ongoing trials, such as Alliance A021806 and PANACHE-PRODIGE 44, have included quality of life as one of the secondary outcomes (Figure S2) [44,48]; their results are required to attain robust conclusions.”
Lines 304–308: “To overcome these drawbacks, systematic reviews and meta-analyses should be performed with a relatively similar patient population (such as in terms of cancer resectability and the AJCC stage); furthermore, comprehensive outcomes encompassing efficacy and quality-of-life assessments should be reported.”
- The conclusion does not actually give any information apart from the gold statement comment on borderline resectable. In view of the conflicting data on NAC in resectable tumours, perhaps the lines from 248-251 ("Overall, although NAT is a novel and promising treatment strategy for resectable PC, both its efficacy and optimal regimen are yet to be determined") should be repeated or placed there as it provides the most pertinent summary
Response: Thank you for this suggestion. Accordingly, we have revised the Conclusion section by repeating the suggested statement at the beginning as shown below (lines 310–311).
“In summary, while NAT shows promise as a novel treatment strategy for resectable PC, its efficacy and optimal regimen are not yet fully elucidated.”

Round 2
Reviewer 1 Report
Comments and Suggestions for Authors
The manuscript by Endo et al. provides a thorough overview of evidence from both published (n=14) and ongoing (n=12) randomized Phase II and III trials of neoadjuvant therapy.
The authors have added flow diagrams of some representative trials (namely Alliance A021806 and PANACHE-PRODIGE 44) as supplementary figures (Figure S2). They are fine.
Reviewer 2 Report
Comments and Suggestions for Authors
authors have met my comments to my satisfaction and the pape ris suitable for publication